# Polymeric nature of tandemly repeated genes enhances assembly of constitutive heterochromatin in fission yeast

Tetsuya Yamamoto [1✉], Takahiro Asanuma[2] & Yota Murakami[2]

Motivated by our recent experiments that demonstrate that the tandemly repeated genes become heterochromatin, here we show a theory of heterochromatin assembly by taking into account the connectivity of these genes along the chromatin in the kinetic equations of small RNA production and histone methylation, which are the key biochemical reactions involved in the heterochromatin assembly. Our theory predicts that the polymeric nature of the tandemly repeated genes ensures the steady production of small RNAs because of the stable binding of nascent RNAs produced from the genes to RDRC/Dicers at the surface of nuclear membrane. This theory also predicts that the compaction of the tandemly repeated genes suppresses the production of small RNAs, consistent with our recent experiments. This theory can be extended to the small RNA-dependent gene silencing in higher organisms.

[1] Institute for Chemical Reaction Design and Discovery, Hokkaido University, Sapporo 001-0021 Hokkaido, Japan. [2] Department of Chemistry, Faculty of Science, Hokkaido University, Sapporo 060-0810 Hokkaido, Japan. ✉email: tyamamoto@icredd.hokudai.ac.jp

The chromatin of a differentiated eukaryotic cell forms heterochromatin that coexists with euchromatin. In many cell types, heterochromatin is observed at the vicinity of the nuclear membrane and the nucleolus[1]. The compartmentalization of heterochromatin regions was also demonstrated in Hi-C experiments[2,3]. The genes in a euchromatin are actively expressed, while the genes in a heterochromatin are only rarely expressed. The genomic region of constitutive heterochromatin, such as centromeric and telomeric regions, does not depend on cell types, while facultative heterochromatin can switch to euchromatin and vice versa during development.

Biophysically, heterochromatin has been thought to be assembled by the phase separation of chromatin[4–18]. The constitutive heterochromatin is characterized by the post-translational modification of histone tails, H3K9me2/3, of nucleosomes. HP1 proteins selectively bind to H3K9 methylated histone tails and show liquid-liquid phase separation due to the multi-valent interaction between these proteins. The multi-valent interaction between HP1 proteins bound to nucleosomes with H3K9me2/3 has been thought to be the driving force of the heterochromatin assembly.

Fission yeast is a classical model system that has been used in molecular biology experiments to study the assembly of heterochromatin[19,20]. A fission yeast has three chromosomes, each having a centromeric region of 40 - 110 kbps (which are estimated to be 24 - 67 Kuhn units). The molecular mechanism of the heterochromatin assembly has been revealed in the last decades. The transcription is a rare event in heterochromatin regions. Nevertheless, the transcription is essential because the RNA interference (RNAi) pathway is the main molecular mechanism of the assembly and maintenance of centromeric heterochromatin of fission yeast[21–24]: RITS complexes bind to nascent RNAs and recruit CLRC complexes to methylate the H3K9 of nucleosomes of heterochromatin[25] or RDRC/Dicers to produce small RNAs[26]. Recent experiments suggest that the binding of RITS complexes to nascent RNAs is small RNA-dependent when they methylate the H3K9 of nucleosomes and is H3K9me2/3-dependent when they produce small RNAs[27,28]: the H3K9 methylation and the small RNA production form the positive feedback, mutually enhancing each other. Experiments suggest that RDRC/Dicers are localized at the inner surface of the nuclear membrane in fission yeast[29,30], implying that the small RNA production (and probably H3K9 methylation as well) occur at vicinity of the surface.

Because the H3K9 methylation and small RNA production occur during transcription, one may wonder what will happen if one upregulates the transcription. Indeed, the heterochromatin mark, such as the level of small RNAs, in the centromeric regions is enhanced by the upregulation of transcription[31]. Why does the transcription form heterochromatin in centromeric regions, but not in euchromatin regions? The centromeric regions are composed of repeat sequences that contain many transcription start sites inside them[31]. Asanuma and coworkers thus knocked-in tandemly repeated euchromatic genes in a euchromatic region to mimic this situation and found that these tandemly repeated genes are favorable substrates for RNAi-mediated heterochromatin—repeat-induced RNAi[31]. This implies that the repeat sequence and many transcription start sites are the key genomic signature, at which constitutive heterochromatin is assembled. It is well-known in polymer physics that polymers adhere to a surface much stronger than their monomers due to the connectivity of monomers along the polymer chain[32,33], see Fig. 1. In a similar manner, the polymeric nature of tandemly repeated genes may enhance the binding of nascent RNAs to RDRC/dicers at the surface of the nuclear membrane.

We therefore construct a theoretical model to predict the assembly mechanism of heterochromatin via repeat-induced RNAi. In this model, we take into account the connectivity of tandemly repeated genes and the diffusion of small RNAs in the kinetic equation of the H3K9 methylation and the small RNA production. Our theory predicts that the polymeric nature of the tandemly repeated genes ensures the steady production of small RNAs and H3K9 methylation through the stable binding to RDRC/Dicers at the surface of the nuclear membrane. We take into account the compaction of the H3K9 methylated chromatin due to Swi6/HP1, motivated by our recent experiments that Epe1, a putative H3K9 demethylase in fission yeast[34–36], is required for repeat-induced RNAi[31]. For simplicity, we do not treat the suppression of transcription due to HDAC that keeps hypoacetylation in heterochromatin. This theory predicts that the compaction limits the accessibility of Pol IIs to the tandemly repeated genes and that the depletion of H3K9 demethylase decreases the production rate of small RNAs if the number of genes in the repeat is large enough. Our predictions are consistent with our recent experiments[31]. Small RNA-dependent gene silencing is not limited to fission yeast, but is also found in higher organisms[37–39]. Extensions of our theory may provide insight in the gene silencing mechanism of such systems.

## Results

**Kinetic model of assembly of heterochromatin.** We treat tandemly repeated genes in a long chromosome at the vicinity of the nuclear membrane. We take into account the connectivity of these genes along the chromatin in the kinetic equations of transcription, H3K9 methylation, and small RNA production, coupled with the diffusion equation of small RNAs, see Fig. 2a. This approach is the fusion of systems biology, which often quantifies biochemical reactions by kinetic equations, and polymer physics, which studies the properties arising from the connectivity of repeated units along a polymer chain. Our model predicts the probability $p$ that the nascent RNAs produced from genes during transcription are connected to RDRC/Dicers localized at the surface of the nuclear membrane, which characterizes the extent of small RNA production (the glossary of symbols are given in Supplementary Table 2).

We use a simplified version of the Stasevich model[40] to treat the transcription, see Fig. 2b. With this model, a gene is either of the three states: the unbound state (where Pol IIs are absent from the gene), the bound state (where a Pol II is bound to the promoter of the gene, but has not yet initiated transcription), the elongation state (where the gene is in the middle of the transcription). The kinetic equations of the state transitions are shown in the Materials and Methods (see also Supplementary Table 2 for the glossary of symbols). Because the binding and unbinding of Pol IIs to the promoter is faster than the initiation of transcriptional elongation, the kinetics of transcription can be represented by the Michaelis-Menten kinetics, where Pol IIs act as enzymes and the promoters act as substrates. Both H3K9 methylation and small RNA production happen during the transcriptional elongation. In the steady state, the fraction $n_{elo}$ of genes during the transcriptional elongation thus has the form

$$n_{elo} = \frac{k_{ini}\tau_{elo}\rho}{(1 + k_{ini}\tau_{elo})\rho + K_p} \qquad (1)$$

where $k_{ini}$ is the rate of transition from the bound to elongation states, $\tau_{elo}$ is the elongation time, $\rho$ is the local concentration of Pol II, and $K_p$ is the equilibrium constant that accounts for the binding of Pol IIs to the promoter, see Materials and Methods for the derivation.

In polymer physics, a long polymer, including chromatin, is modeled as repeated units, each has Kuhn length $b$, connected along a chain. The Kuhn length of chromatin is estimated to be

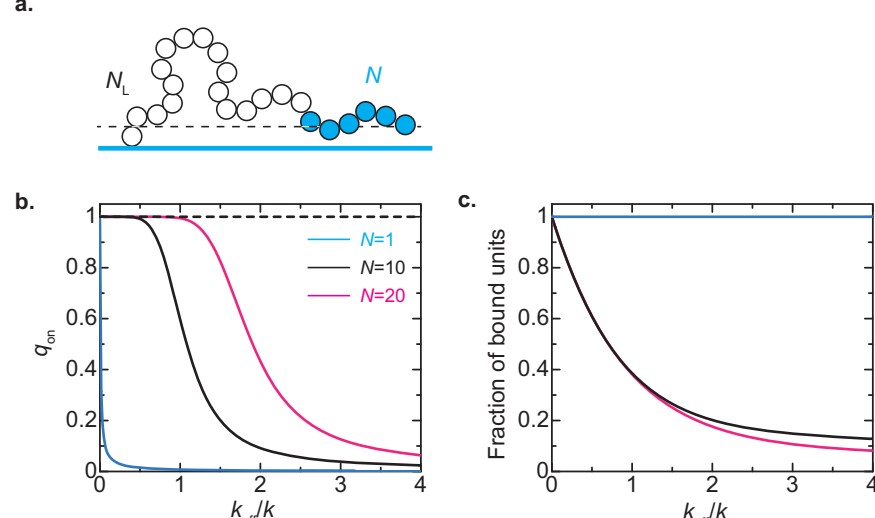

**Fig. 1 Adhesion of polymers to a surface. a** Polymers composed of $N$ adhesive units (cyan beads) and $N_L$ linker units (white beads). Adhesive units bind to the surface with the rate $k_{on}$ if they are located in $0 < z < b$ and bound units are unbound from the surface with the rate $k_{off}$. Unbound units (for both adhesive and linker units) are repelled by the surface. **b** The probability $q_{on}$ that more than one (adhesive) unit is bound to the surface is shown as a function of the binding constant $k_{off}/k_{on}$ for $N = 1$ (cyan), 10 (black), and 20 (magenta). **c** The fraction of units bound to the surface, provided that at least one (adhesive) unit is bound to the surface, is shown as a function of the binding constant $k_{off}/k_{on}$ for $N = 1$ (cyan), 10 (black), and 20 (magenta). **b** and **c** are derived by using the scaling theory shown in Supplementary Note 1, see also Supplementary Figures 1–5 and Supplementary Table 1. We used $N_L = 200$ to derive **b**.

50 nm ($\approx$1.65 kbps)[41], which is approximately the length of typical genes, such as *ade6* ($\approx$1.66 kpbs). We treat such cases, in which each chromatin unit has one gene. Nascent RNAs produced by the transcription of heterochromatin regions are retained to the chromatin via RITS complexes[28,42,43]. A complex of the chromatin unit and nascent RNA can be thus viewed as one unit that can bind to RDRC/Dicers at the surface of the nuclear membrane. The kinetic equation of the fraction $p$ of units bound to RDRC/Dicers thus has the form

$$\frac{dp}{dt} = k_{on} \frac{b}{\xi} \sigma n_{elo}(1 - p) - \frac{p}{\tau_{elo}} \qquad (2)$$

see also Fig. 2a. The first term of Eq. (2) is the rate of the binding of chromatin units to RDRC/Dicers at the surface and the second term represents the fact that bound units are released from RDRC/Dicers at the termination of transcription. The binding of a unit to RDRC/Dicers happens only if the nucleosomes of the unit are H3K9 methylated[28] and is during the transcriptional elongation. The first term of Eq. (2) is thus proportional to the fraction $\sigma$ of H3K9 methylated nucleosomes and the probability $n_{elo}$ that this unit is in the elongation state. Polymer physics has revealed that polymer units are confined in a layer of thickness $\xi$, which is the size of a subchain between units bound to the surface[32,33], see also Supplementary Note 1. The polymeric nature of the tandemly repeated genes is taken into account in the factor $b/\xi$ in the first term of Eq. (2). We use the simplest approximation in polymer physics—ideal chain approximation, to derive the expression of the size $\xi$ of the subchain in the next section, see Fig. 3a, and take into account the compaction of the subchain due to the attractive interaction between Swi6 in the following sections, see Fig. 3b.

Small RNAs are produced by RDRC/Dicers that cut nascent RNAs into pieces. The rate of the production of small RNAs per unit area has the form

$$S = \frac{s_0 N p}{1 - (1 - p)^N} \frac{1}{\xi_N^2} \qquad (3)$$

see also Fig. 2a and Supplementary equation (S35). $s_0$ is the production rate of small RNAs from a unit. The production rate $S$ per unit area is proportional to the number of units that are bound to RDRC/Dicers and to the inverse of the area $\xi_N^2$ occupied by the tandemly repeated genes. The produced small RNAs diffuse away from the nuclear membrane. In the steady state, the local concentration of small RNAs has the form

$$c(z) = \frac{S}{k_d \lambda} e^{-\frac{z}{\lambda}}. \qquad (4)$$

Equation (4) represents the fact that small RNAs are localized at the region of thickness $\lambda$. The diffusion length $\lambda (= \sqrt{D/k_d})$ is the distance by which small RNAs can diffuse (with the diffusivity $D$) until they are degraded (with the rate $k_d$). While small RNAs diffuse away from the nuclear membrane, they are bound to (and unbound from) argonaute proteins in RITS complexes. The concentration $c(z)$ includes both bound and unbound populations of small RNAs.

The kinetic equation of the degree $\sigma$ of H3K9 methylation has the form

$$\frac{d\sigma}{dt} = k_m \Lambda n_{elo}(1 - \sigma) - k_{dm}\sigma \qquad (5)$$

see Fig. 2a. Equation (5) represents the fact that the degree of H3K9 methylation is determined by the balance of H3K9 methylation rate (the first term) and H3K9 demethylation rate (the second term). Equation (5) also assumes that the number of CLRC and RITS complexes is large and does not limit the kinetics of H3K9 methylation and that H3 proteins in nucleosomes are stably bound to DNA and are not evicted during the transcription, reflecting the recent structural biology experiments[44]. The parameter $\Lambda$ is proportional to the probability that small RNAs that are bound to RITS complexes are bound to the nascent RNAs produced from a gene,

$$\Lambda \simeq \frac{1}{\xi} \int_0^\xi dz c(z) = \frac{S}{k_d \xi} \left(1 - e^{-\frac{\xi}{\lambda}}\right) \qquad (6)$$

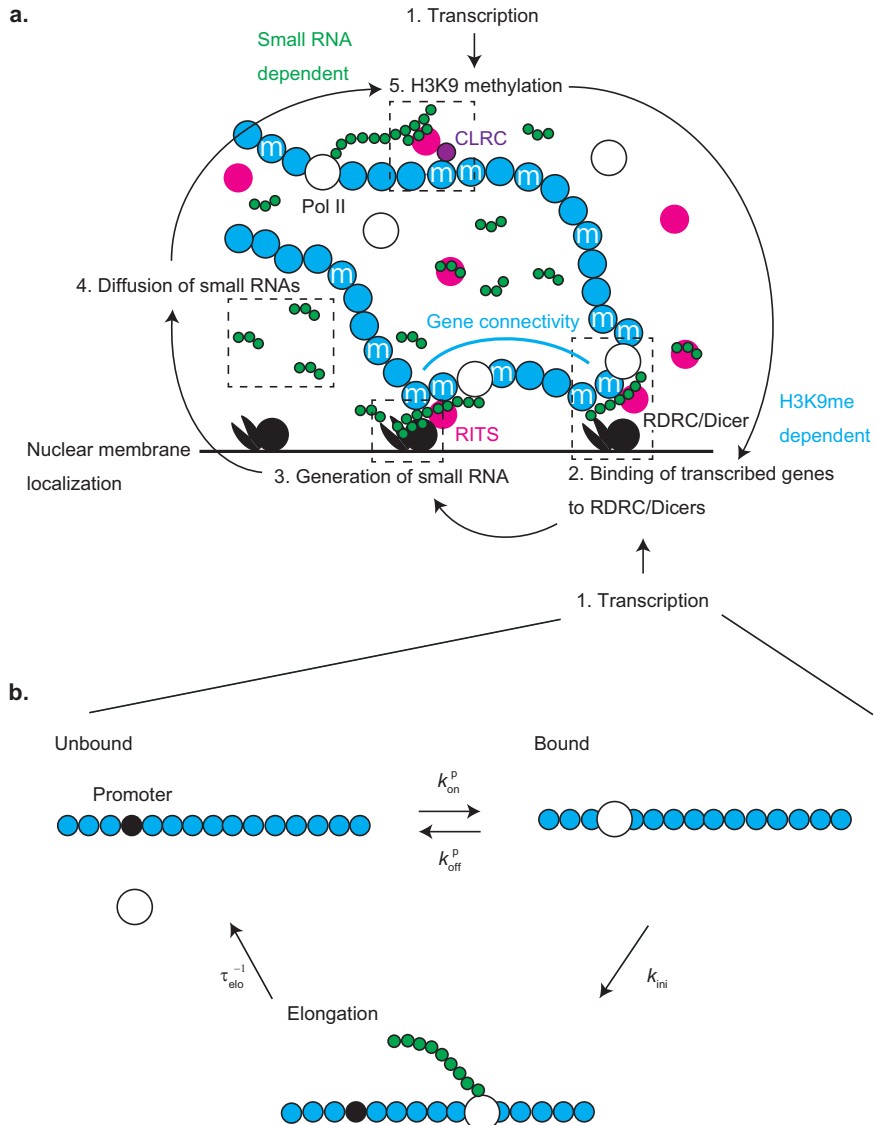

**Fig. 2 Model of heterochromatin assembly. a** The key processes of heterochromatin assembly taken into account in our model are transcription (1), the binding of nascent RNAs (long green chain) to RDRC/Dicers (black complex) at the surface of the nuclear membrane (2), the production of small RNAs (short green chain) (3), the diffusion of small RNAs (4), and the H3K9 methylation of nucleosomes (cyan circles) (5). The production of small RNAs is H3K9me-dependent and the H3K9 methylation is small RNA-dependent. These processes thus mutually enhance to each other (positive feedback). Nascent RNAs are necessary for the binding of nascent RNAs to RDRC/Dicers and H3K9 methylation. The latter processes happen during transcriptional elongation. RITS complexes (magenta circle) act as a hub that mediate the binding of nascent RNAs and RDRC/Dicers or CLRCs (purple), but are taken into account in the model only implicitly. If more than one gene are bound to a RDRC/Dicer, unbound genes in the tandem repeat are also localized at the vicinity of RDRC/Dicers because the genes are connected through the chromatin and RDRC/Dicers are localized at the surface of the nuclear membrane, see the thick cyan and black lines. **b** Each gene is either of the three state: Pol II are absent from the gene (unbound), Pol II is bound to the promoter (bound), and Pol II has engaged in the transcriptional elongation (elongation). The transition between the unbound and bound states is faster than the transition from the bound to elongation states.

see also Fig. 2a. To derive Eq. (6), we assumed that the dynamics of the subchain is faster than the enzymatic reaction of the H3K9 methylation.

The binding probability $p$ of units is derived by solving Eqs. (1–6) for the steady state and the relationship between the size $\xi$ of the tandemly repeated genes and the probability $p$ that depend on polymer models (shown in the following sections). This conditional probability is effective for the bound state, at which at least one unit in the repeat is bound to RDRC/Dicers at the surface of nuclear membrane. The probability $q_{on}$ that the repeat

is in the bound state follows the kinetic equation

$$\frac{d}{dt}q_{on} = k_{on}\frac{3}{2}\frac{N}{N_L}\sigma_0(1-q_{on}) - N\frac{p(1-p)^{N-1}}{1-(1-p)^N}\frac{q_{on}}{\tau_{elo}} \quad (7)$$

Equation (7) represents the fact that the probability $q_{on}$ increases if one of the units in unbound tandemly repeated genes is bound to a RDRC/Dicer (the first term) and decreases if the transcription of the gene at the last bound unit is terminated before other genes are bound to RDRC/Dicers (the second term).

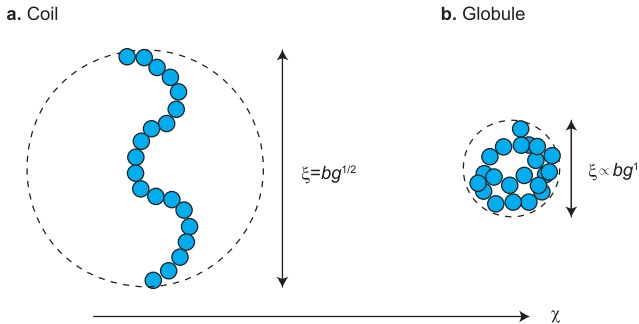

**Fig. 3 Coil-globule transition.** If we neglect the interaction between units (ideal chain approximation), the size of a section composed of $g$ units in a chromatin is $\xi = bg^{1/2}$ (coil) (**a**). In general, the H3K9 methylated units in the section show attractive interaction via Swi6. The interaction parameter $\chi$ represents the magnitude of this interaction. If the interaction parameter is large enough, the size of the section is $\xi \propto bg^{1/3}$ (globule) (**b**).

$\sigma_0$ is the degree of H3K9 methylation of the tandemly repeated genes due to the primary small RNA for cases that all of the genes are not bound to RDRC/Dicers. $N_L$ is the number of units of the linker chromatin (between the tandemly repeated genes and the bound chromatin region at the neighbor, such as telomere), see Fig. 1a. To derive the second term, we assumed that the relaxation time for the binding of units in the tandemly repeated genes is shorter than the time scale of the kinetics of $q_{on}$ and that the unbinding of units results from the termination of transcription (a more general derivation of $q_{on}$ is shown in sec. S1.4 and S1.6 in the SI).

**Polymeric nature of tandemly repeated genes ensures steady production of small RNAs.** The tandemly repeated genes are confined in a layer of thickness $\xi$ of the size of a subchain between units bound to RDRC/Dicers at the surface of the nuclear membrane. Because $pN$ units are bound to the surface, the average number of units in a subchain is $p^{-1}$. The simplest treatment of the polymeric nature of the tandemly repeated genes is the ideal chain approximation that takes into account only the connectivity of the genes[32]. With this approximation, its size $\xi$ has the form

$$\xi = bp^{-\frac{1}{2}}, \tag{8}$$

see also Fig. 3a and Supplementary Note 1. $\xi_N$ in Eq. (3) is the size of the tandemly repeated genes (composed of $N$ units) and is $bN^{1/2}$ in the ideal chain approximation. We also neglect the excluded volume interaction between chromatin units and Pol IIs. The local concentration $\rho$ of Pol II at the genes is thus equal to the concentration $\rho_0$ of Pol II in the nucleosol, $\rho = \rho_0$. The values of the parameters used for the numerical calculations are summarized in Table 1. The general feature of our results does not depend on these values, see also Supplementary Figures 6–8.

The binding probability $p$ is derived as a function of the inverse of the (rescaled) elongation time $1/(k_{on} \tau_{elo})$, see Fig. 4. The latter dimensionless quantity corresponds to the (rescaled) dissociation rate $k_{off}/k_{on}$ in the polymer adhesion problem, see Fig. 1b. For cases in which the number of genes in the repeat is smaller than a critical value $N_c$, the binding probability $p$ increases continuously with increasing the elongation time $\tau_{elo}$, see the cyan line in Fig. 4. In contrast, for cases in which the number of genes in the repeat is larger than the critical value $N_c$, the binding probability $p$ has two stable solutions (the magenta solid line Fig. 4) and one unstable solution (the magenta broken line in Fig. 4). The binding probability $p$ is approximately zero for one of the two solutions, implying that the units in the repeat are rarely bound to RDRC/

Dicers and do not produce small RNAs steadily. This feature is analogous to euchromatin. The binding probability $p$ is relatively large in the other stable solution, implying that the units in the tandemly repeated genes are almost always bound to RDRC/Dicers and steadily produce small RNAs. This feature is analogous to heterochromatin. The euchromatin solution is stable for $\tau_{elo}^{-1} > \tau_{sp2}^{-1}$, while the heterochromatin solution is stable for $\tau_{elo}^{-1} < \tau_{sp1}^{-1}$, see the magenta line in Fig. 4. The probability $p$ therefore jumps from zero to a finite value at $\tau_{elo} = \tau_{sp2}$ and from a finite value to zero at $\tau_{elo} = \tau_{sp1}$. Our theory therefore predicts the discontinuous transition (the first order phase transition) between euchromatin and heterochromatin. Another important prediction is that not only the number of genes in the repeat, but also the elongation time, which is the length of each gene divided by the elongation of Pol II, are critical parameters for the assembly of heterochromatin. The latter prediction may be experimentally accessible.

The probability $q_{on}$ of the repeat at the bound state increases as the elongation time $\tau_{elo}$ increases and eventually becomes $q_{on} \approx 1$, where the repeat is stably bound to the surface of nuclear membranes, see Fig. 5a. Notably, the width of the window of elongation time at which $q_{on} \approx 1$ increases as the number $N$ of genes increases. It is because the rate with which all the genes in the repeat are unbound from RDRC/Dicers (shown in the second term of Eq. (7)) decreases exponentially with increasing number $N$ of genes. This mechanism is essentially the same as the case of the surface adhesion of polymers, see Fig. 1b, c. The average degree $\sigma q_{on}$ of H3K9 methylation and the average production rate of small RNAs increase with increasing the number $N$ of genes and eventually saturate for the regime of long elongation time, reflecting the feature of $q_{on}$, see Fig. 5b, c. The increase of the level of H3K9 methylation and small RNAs with the number $N$ of genes in the tandem repeat is consistent with our recent experiments[31]. Our theory therefore predicts that the polymeric nature of the tandemly repeated genes enhances the positive feedback between the small RNA production and the H3K9 methylation through the stable binding to RDRC/Dicers at the surface of the nuclear membrane, ensuring the steady production of small RNAs. Yet, the binding probability $p$ in the bound state does not increase with $N$ because although the production rate of small RNAs increases in proportion to the number $N$ of genes in the repeat, the produced small RNAs are diluted due to the fact that the area $\xi_N^2$ occupied by this repeat increases in proportion to the number $N$.

**Chromatin compaction due to Swi6/HP1 suppresses transcription and reduces the production of small RNA.** Swi6/HP1 binds to H3K9 methylated nucleosomes[45] and the self-association of Swi6 has been thought to promote the compaction of the centromeric regions[46–49]. The extent of the chromatin compaction is quantified by the volume fraction $\phi_c$ of chromatin. The volume fraction $\phi_c$ is determined by the equation of state

$$-p^{\frac{4}{3}}\phi_c^{\frac{1}{3}} + p^{\frac{2}{3}}\phi_c^{\frac{5}{3}} - \chi\sigma^2\phi_c^2 - \chi_0\phi_c^2 - \phi_c - \log(1-\phi_c) = 0. \tag{9}$$

The size $\xi$ of a subchain is derived from the expression of the volume fraction $\phi_c = b^3 p^{-1}/\xi^3$. Equation (9) represents the balance of the elastic stress due to the entropic elasticity of chromatin (the first and second terms), the stress due to the attractive interaction between chromatin units via Swi6 (the third term), and the stress due to the excluded volume interaction between chromatin units (the fourth, fifth, and sixth terms). Equation (9) is an extension of the equation of state that predicts the coil-globule transition of polymers and is derived by using the free energy in the spirit of the Flory theory[50], see also Fig. 3 and

**Table 1 Independent parameters of the ideal chain model.**

| Parameters | Meaning | Values |
|---|---|---|
| $\frac{k_{\text{ini}}}{k_{\text{on}}}\frac{\rho_0}{\rho_0+K_p}$ | Rescaled transcription rate | 0.2 |
| $\lambda/b$ | Diffusion length | 10 |
| $\frac{s_0}{b^2\sqrt{Dk_d}}\frac{k_m}{k_{\text{dm}}}$ | Production rate | 0.1 |
| $N_L/\sigma_0$ | Linker chromatin length | $2\times10^4$ |

The values of independent parameters used for our numerical calculations are summarized. The binding constant $K_p/\rho_0$ ($\approx7.7$) and the initiation rate $k_{\text{ini}}$ ($\approx0.005\,\text{s}^{-1}$)) were estimated from experiments on a mouse adenocarcinoma cell line[40]. The binding rate $k_{\text{on}}$ is not available in the literature, but we estimated to be a similar order to $k_{\text{ini}}$. The diffusivity $D$ of small RNAs (of length $b_s\approx20$ nt $\approx6$ nm) is estimated as $D\approx\frac{k_BT}{4\pi\eta_wb_s}\approx55\,\mu\text{m}^2/\text{s}$ ($\eta_w$ is the viscosity of water, $k_B$ is the Boltzmann constant, and $T$ is the absolute temperature). The degradation rate $k_d$ and production rate $s_0$ of small RNAs as well as the kinetic constants, $k_m$ and $k_{\text{dm}}$, for H3K9 methylation and demethylation were not available in the literature. The number $N_L$ of units in the linker chromatin is estimated as 200, which corresponds to $\approx330$ kb. The degree $\sigma_0$ of H3K9 methylation solely due to the primary small RNAs was also not available in the literature and is estimated to be $\approx0.01$. The number $N$ of genes in the repeat and the rescaled elongation time $k_{\text{on}}\tau_{\text{elo}}$ were changed systematically.

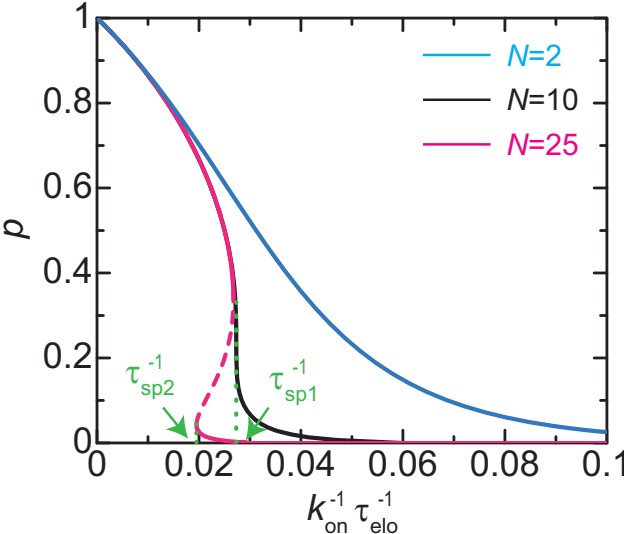

**Fig. 4 Binding probability $p$ vs elongation time $\tau_{\text{elo}}$ – ideal chain chromatin model.** The binding probability $p$ is shown as a function of the inverse of the elongation time $\tau_{\text{elo}}$ as predicted by using the ideal chain chromatin model, see Eq. (8). We performed numerical calculations for $N=2$ (cyan), 10 (black), and 25 (magenta). The stable solutions are shown by the solid lines and the unstable solution is shown by the broken line. $\tau_{\text{sp1}}$ and $\tau_{\text{sp2}}$ are the values of the elongation time at which the heterochromatin and euchromatin solutions become unstable, respectively, see the light green dotted lines. The parameters used for the calculations are summarized in Table 1. The derivation of this figure, including the critical number $N_c$ of genes, is shown in Supplementary Note 2.

Eq. (17) in Materials and Methods (the detail of the derivation is shown in Supplementary Note 3). This treatment is effective for cases in which the binding and unbinding of the units is rate limited: in such cases, the polymer dynamics associated with the binding and unbinding of the units is negligible and thus the local equilibrium approximation is applicable in the length scale of the subchain. $\chi$ is the interaction parameter that represents the magnitudes of the attractive interaction between chromatin units via HP1, see the dependence of the third term on $\sigma^2$. $\chi_0$ is the interaction parameter that represents the magnitudes of the interaction independent of Swi6. We here use $\chi_0=\frac{1}{2}$ to approximately treat the screening of the excluded volume interaction due to the overlapping of chains: the chromatin return to ideal chains in the limit of vanishing Swi6 interaction, $\chi$, see Eq. (3) (because the first and second terms of Eq. (9) dominate the other terms in this limit). The treatment used in this section is thus a direct extension of the treatment used to derive Fig. 1 (which agrees with the molecular dynamics simulation, see Supplementary Note 1).

In the globular state, the subchains composed of $p^{-1}$ units in the tandemly repeated genes do not interpenetrate each other. The area occupied by the tandemly repeated genes is thus $\xi_N^2=\xi^2pN$. This relationship is also valid for the coil state (see also the discussion below Eq. (8)) and will be used for any values of $\chi$ and $\chi_0$. The local concentration $\rho$ of Pol II at the genes has the form

$$\rho=(1-\phi_c)\rho_0 \tag{10}$$

which reflects the excluded volume interaction between Pol IIs and chromatin. The values of the parameters used for the numerical calculations are summarized in Table 2. The general feature of our results does not depend on these values, see Supplementary Figures 9–14. In the following, we mainly focus on the feature of the bound state of tandem repeats composed of many genes, where it does not depend on the number $N_L$ of units in linker chromatin.

The interaction parameter $\chi$ is a standard parameter that is widely used in polymer physics, but its value for the chromatin interaction via Swi6 is not available in the literature. We thus calculated the binding probability $p$ of units for various values of the interaction parameter $\chi$. Our theory predicts that the binding probability $p$ decreases as the interaction parameter $\chi$ increases for a relatively large elongation time, see Fig. 6a. This implies that the chromatin compaction due to Swi6 indeed suppresses the assembly of heterochromatin. Indeed, the production rate $S\xi_N^2$ of

small RNAs and the degree $\sigma$ of H3K9 methylation in the bound state decreases with increasing the interaction parameter $\chi$ for the regime of long elongation time, see Fig. 6b, c. It is because the chromatin compaction decreases the local concentration of Pol IIs at the tandemly repeated genes and thus decreases the transcription rate of these genes, see Fig. 6d and Eq. (10).

**H3K9 loss enhances the production of small RNA if the number of repeated genes is large.** Epe1 is a putative H3K9 demethylase in fission yeast[35,36]. We have experimentally demonstrated that the depletion of Epe1 significantly decreases the production of small RNAs[31]. In our theory, the action of Epe1 is implicitly taken into account through the H3K9 demethylation rate $k_{\text{dm}}$. We thus analyzed the dependence of the features of the heterochromatin on the demethylation rate $k_{\text{dm}}$, see Fig. 7. Our theory predicts that the binding probability $p$ of units is a non-monotonic function of the demethylation rate $k_{\text{dm}}$ if the interaction parameter is large enough, see the orange and magenta lines in Fig. 7a. It implies that if the inverse demethylation rate $k_{\text{dm}}^{-1}$ is larger than the maximum in the wild type, the binding probability $p$ decreases with decreasing the rate $k_{\text{dm}}$ (which corresponds to the depletion of Epe1). In this regime, the production rate of small RNAs in the bound state also decreases with decreasing the demethylation rate $k_{\text{dm}}$, while the degree of H3K9 methylation increases with decreasing the demethylation rate $k_{\text{dm}}$, see the orange and magenta lines in Fig. 7b, c. The down-regulation of small RNA production by the depletion of Epe1 results from the fact that the local concentration $\rho$ of Pol II in the tandemly repeated genes decreases because this chromatin region becomes more compact with the depletion of Epe1, see the orange and magenta lines in Fig. 7d.

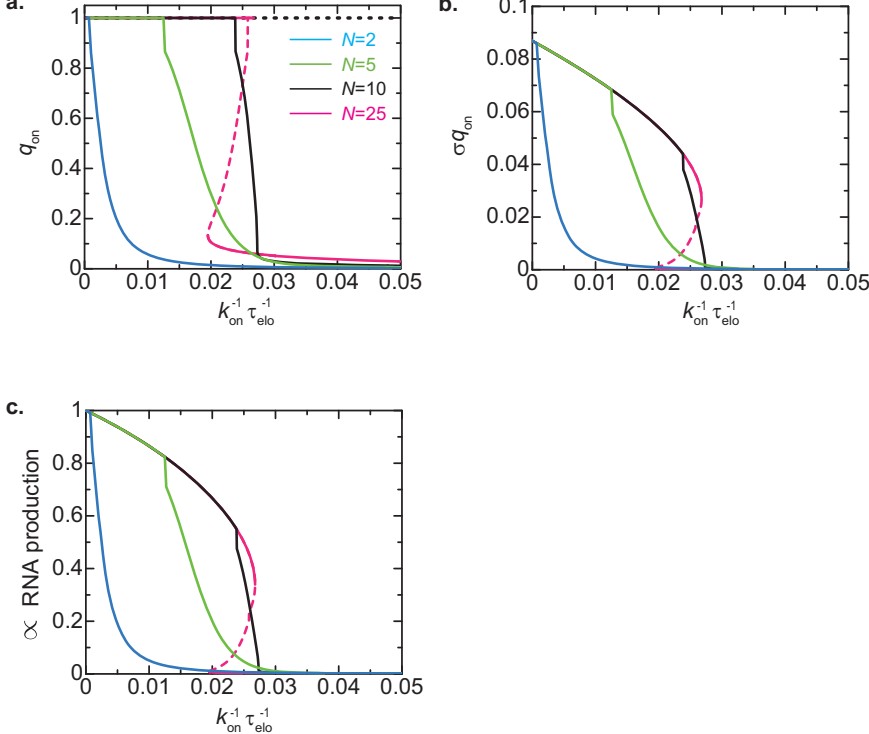

**Fig. 5 Degree of H3K9 methylation and production rate of small RNAs vs elongation time – ideal chain chromatin model.** The probability $q_{on}$ that at least one unit in the tandemly repeated genes is bound to RDRC/Dicers (**a**), the degree $\sigma q_{on}$ of DNA methylation (**b**), and the rescaled production rate $pq_{on}/(1-(1-p)^N)$ of small RNAs (**c**) are shown as functions of the inverse of the elongation time $\tau_{elo}$ as predicted by the ideal chain chromatin model. We performed numerical calculations for $N = 2$ (cyan), 5 (light green), 10 (black), and 25 (magenta). The values of parameters used for the calculations are summarized in Table 1.

Our theory takes into account the primary small RNAs through the degree $\sigma_0$ of H3K9 methylation in the unbound state, see Eq. (7). The degree $\sigma_0$ increases with the depletion of H3K9 demethylase. For the case of only a gene of single copy ($N = 1$), the H3K9 methylation is mainly due to the primary small RNAs and the degree of H3K9me thus increases with the depletion of H3K9 demethylase. The H3K9 methylation of tandemly repeated genes is mainly due to the secondary small RNAs and the degree of H3K9me increases with the depletion of H3K9 demethylase, as described in the preceding section.

**Transcription upregulation increases both RNA production and H3K9 methylation.** The idea of chromatin compaction dependent gene accessibility of Pol II is attractive because the contribution of Epe1 to the heterochromatin assembly is explained only by the putative role of Epe1 as H3K9 demethylase. Epe1 has not only a JmjC domain, which is the putative domain of H3K9 demethylation activity, but also a transcription activation domain[51]. The relationship between the H3K9 demethylation and transcriptional activation is not fully elucidated. We first assume that the dominant role of epe1 is the activation of transcription, rather than the H3K9methyaltion, and use the ideal chain model to analyze if this assumption is consistent with our experimental results. In our theory, the transcriptional activator increases the rescaled transcription rate constant $\frac{k_{ini}}{k_{on}}\frac{\rho_0}{\rho_0+K_p}$, see Fig. 2 and Table 1. Our theory predicts that both the small RNA production rate and the degree $\sigma$ of H3K9 methylation decrease with decreasing the transcription rate constant, namely with the depletion of transcriptional activator, if the latter constant is larger than the value at the euchromatin-heterochromatin transition, see the solid lines in Fig. 8a, b. Our recent experiments

suggest that the level of H3K9 methylation increases with the depletion of Epe1[31], implying that these experimental results are not explained only by the role of Epe1 as a transcription activator.

Both of the rescaled transcription rate constant $\frac{k_{ini}}{k_{on}}\frac{\rho_0}{\rho_0+K_p}$ and the demethylation rate constant $k_{dm}$ may decrease with the depletion of Epe1 due to its dual role as a H3K9 demethylase and a transcriptional activator. Both the production rate of small RNAs and the degree $\sigma$ of H3K9 methylation increase with decreasing the demethylation rate $k_{dm}$, see the black and red lines in Fig. 8a, b. This implies that whether the dual role of Epe1 is consistent with our experiments depends on the details of the modulation of the kinetic parameters by Epe1 depletion, which have not been well-characterized.

The analytical forms may be useful for future experiments because they represent the explicit dependence on the rate constants. For $p \approx 1$, $\sigma < 1/2$, and $N \gg 1$, the binding probability $p$, which is proportional to the production rate of small RNAs, has an approximate form

$$p = 1 - \frac{1}{k_{on}\tau_{elo}}\frac{b^2\sqrt{Dk_d}}{s_0}\frac{k_{dm}}{k_m}\frac{1}{n_{elo}^2} \qquad (11)$$

where it depends on the transcription rate constant via $n_{elo}$ given by Eq. (1), see the dotted lines in Fig. 8a. The degree $\sigma$ of H3K9 methylation has an approximate form

$$\sigma = \frac{s_0}{b^2\sqrt{Dk_d}}\frac{k_m}{k_{dm}}n_{elo} - \frac{1}{k_{on}\tau_{elo}n_{elo}} \qquad (12)$$

see the dotted lines in Fig. 8b. The binding probability $p$ and the degree $\sigma$ of H3K9 methylation are sensitive to the transcription rate constant at the vicinity of the euchromatin-heterochromatin transition.

## Discussion

We have constructed a theory of the assembly of the constitutive heterochromatin in fission yeast. This theory is motivated by the fact that the repeat sequence and many transcription start sites are the genomic signature of heterochromatin assembly in fission yeast[31]. It is analogous to the surface adhesion of polymers[32,33], given the fact that small RNAs are produced by RDRC/Dicers at the surfaces of the nuclear membrane[29,30]. We thus take into account the essence of the surface adhesion of polymers in the kinetic equations of the small RNA production and the H3K9 methylation. Quantifying biochemical reactions by using the kinetic equations is an approach often used in systems biology, while the properties arising from the connectivity of repeated units along a chain have been studied in polymer physics: our approach is the fusion of the soft matter physics and the systems biology. It is very different from other theories of heterochromatin assembly, which are based on the theory of phase separation[4–16].

Our theory predicts that the polymeric nature of the tandemly repeated genes enhances the positive feedback between the small RNA production and the H3K9 methylation through the stable binding to RDRC/Dicers at the surface of the nuclear membrane, ensuring the steady production of small RNAs. The stable binding is promoted by the localization of nascent RNAs along DNA and of RDRC/Dicers on the surface of the nuclear membrane with a mechanism analogous to the surface adhesion of polymers: if a nascent RNA produced from a gene is bound to a RDRC/Dicer, other nascent RNAs in the tandem repeat are at the vicinity to the surface, where other RDRC/Dicers are localized. This is a new insight over the model proposed in our recent experimental paper[31] that the tandem repeat increases the local concentration of nascent RNAs. This mechanism also acts to the cases in which the sequence is not a sequence of exact repeat, but a sequence including many transcription start sites, such as the case of dg/dh in centromeric region of fission yeast, which can produce transcripts containing the same sequence. We note that the fact that RDRC/Dicers are localized at the surface of a nuclear membrane plays a similar role in the localization of RDRC/Dicers(/RITS) in a condensate assembled by phase separation. A similar principle may thus act in a completely different system, such as the stable binding of tandemly repeated rDNAs to the surfaces of subcompartments in nucleoli[52–54] and the stable binding of tandemly repeated enhancers in a superenhancer to the surfaces of transcriptional condensate[55]. Our theory also predicts that not only the number of genes in the repeat, but also the elongation time, which is the length of each gene divided by the elongation rate of

**Table 2 Independent parameters of the coil-globule transition model.**

| Parameter | Meaning | Values |
|---|---|---|
| $\rho_0$ | Pol II volume fraction | 0.12 |
| $\dfrac{k_{ini}}{k_{on}}\dfrac{1}{K_p}$ | Transcription initiation rate | 1.48 |
| $\lambda/b$ | Diffusion length | 10.0 |
| $\dfrac{s_0}{b^2\sqrt{Dk_d}}\dfrac{k_m}{k_{dm}}$ | Production rate | 2.0 |
| $\chi$ | Interaction parameter (HP1) | Varied |
| $\chi_0$ | Interaction parameter (screening) | 0.5 |

The volume and concentration of Pol II are estimated to be $6 \times 10^3$ nm$^3$ (see ref. [62] and μM (the number of Pol II per cell is $3 \times 10^4$ (see ref. [63]) and the size of a nucleus of fission yeast is in the order of 1 μm$^3$ (see ref. [64]), respectively. $k_{ini}/(k_{on} K_p)$ was determined to be consistent with Table 1. The interaction parameter $\chi_0$ is set to 0.5 to represent the screening of the excluded volume interaction. The parameter $\chi$ for the chromatin interaction via HP1 is systematically varied. See also the caption of Table 1.

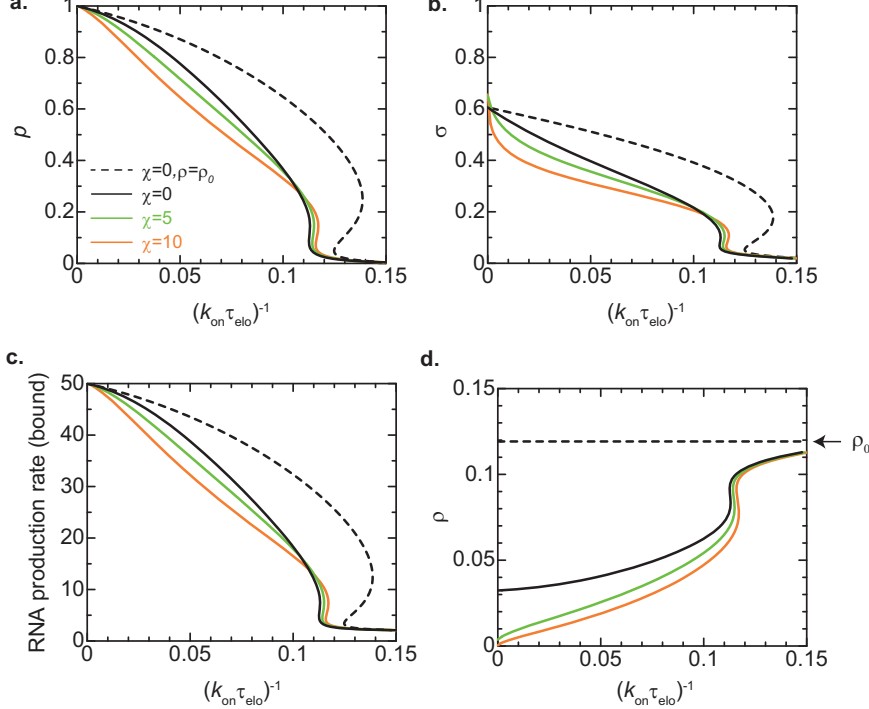

**Fig. 6 Effect of chromatin compaction on heterochromatin assembly – coil globule transition model.** The binding probability $p$ (**a**), the degree $\sigma$ of H3K9 methylation in the bound state (**b**), the RNA production rate $S\xi_N^2$ in the bound state (**c**), and the volume fraction $\rho$ of Pol II (**d**) are shown as functions of the inverse of the elongation time for the values of the interaction parameters $\chi = 0.0$ (black), 5.0 (light green), and 10.0 (orange) as predicted by using the coil-globule transition model. The black broken line is derived by using Eq. (9) with $\rho = \rho_0$. The number $N$ of units in the tandemly repeated genes is set to 25. The values of other parameters used for the calculations are summarized in Table 2.

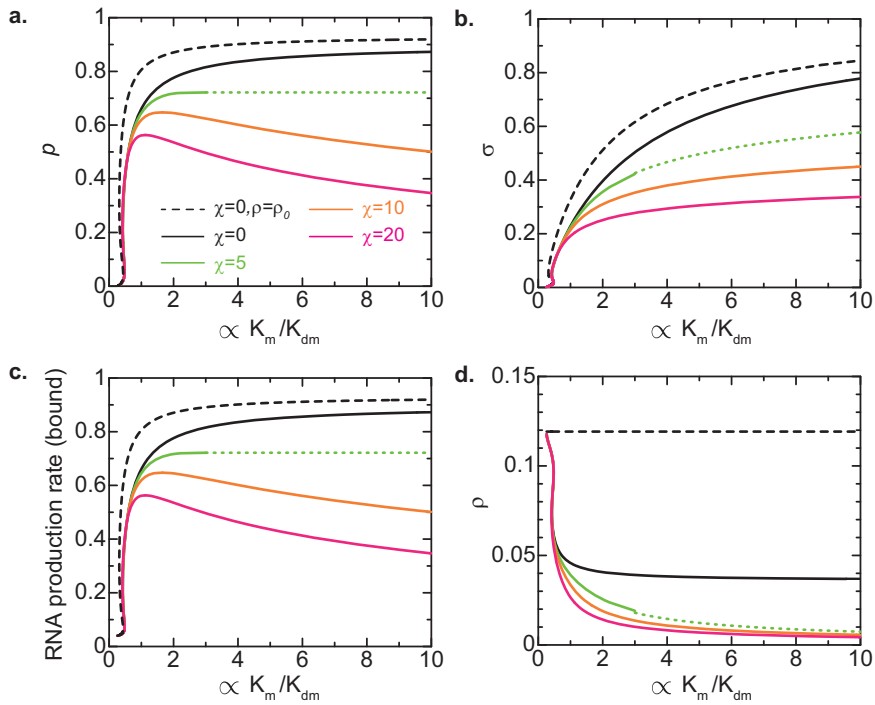

**Fig. 7 Effect of demethylase upregulation on heterochromatin assembly – coil-globule transition model.** The binding probability (**a**), the degree $\sigma$ of H3K9 methylation in the bound state (**b**), the rescaled RNA production rate $p/(1-(1-p)^N)$ in the bound state (**c**), and the volume fraction of Pol II (**d**) are shown as functions of $\frac{s_0}{b^2\sqrt{Dk_d}}\frac{k_m}{k_{dm}}$ (which is proportional to the inverse of the H3K9 demethylation rate $k_{dm}$) for $\chi = 0$ (black), 5.0 (light green), 10.0 (orange), and 20.0 (magenta) as predicted by the coil-globule transition model. The black broken line is derived by using Eq. (9) with $\rho = \rho_0$. The inverse of rescaled elongation time $(k_{on}\tau_{elo})^{-1}$ is set to 0.05. The green dotted lines are derived by using the value of the binding probability for $\frac{s_0}{b^2\sqrt{Dk_d}}\frac{k_m}{k_{dm}} \to \infty$. The number $N$ of units in the tandemly repeated genes is set to 25. The values of other parameters used for the calculations are summarized in Table 2.

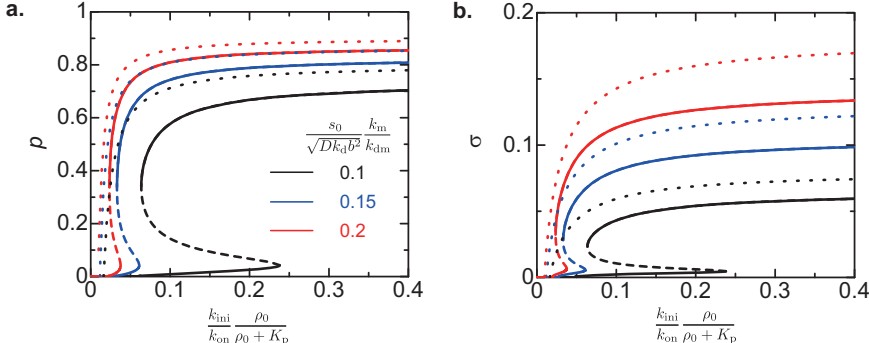

**Fig. 8 Dependence of small RNA production and H3K9 methylation on transcription rate – ideal chain model.** The binding probability $p$, which is proportional to the production rate of small RNAs, (**a**) and the degree $\sigma$ of H3K9 methylation (**b**) are shown as functions of the rescaled transcription rate constant $\frac{k_{ini}}{k_{on}}\frac{\rho_0}{\rho_0+K_p}$ for the inverse of the rescaled H3K9 demethylation rate $\frac{s_0}{\sqrt{Dk_d}b^2}\frac{k_m}{k_{dm}} = 0.1$ (black), 0.15 (blue), and 0.2 (red), as predicted by the ideal chain model. The solid line is derived by the numerical calculation and the dotted lines are derived by using Eqs. (11) and (12). The inverse of elongation time $1/(k_{on}\tau_{elo})$ is fixed to 0.02. Other parameters are shown in Table 1.

Pol II, are important parameters to the assembly of heterochromatin, see Fig. 5. The latter prediction may be accessible by measuring the dependence of the levels of small RNAs and H3K9 methylation on the length of each gene in the repeat. The dependence of the heterochromatin assembly on the elongation rate is not because of a specific regulation in transcription[56], but because a long elongation time increases the chance for nascent RNAs to bind to RDRC/Dicers and CLRC.

We also studied the contribution of the chromatin compaction due to the self-association of Swi6 to the small RNA production

and the H3K9 methylation, see Fig. 6. While it is an issue separate from the nature of the heterochromatin assembly as the polymer surface adhesion, it can regulate the conformation of the tandemly repeated genes and the accessibility of Pol II to these genes. The chromatin compaction mainly affects the local concentration of Pol II at the vicinity of the genes, see Eq. (10). Our theory predicts that the production of small RNAs decreases as the H3K9 demethylation rate decreases if the number of genes in the repeat is large because it decreases the local concentration of Pol II. This prediction is consistent with our experimental result on

the depletion of a putative H3K9 demethylase, Epe1[31]. This does not exclude the possibility that the dual effect of Epe1 on H3K9 demethylation and transcriptional activation, but not via the chromatin compaction, can explain the experimental result, see Fig. 8. However, it depends on the details of the modulation of the kinetic parameters by Epe1, which is beyond the scope of this paper. In both cases, it is necessary to take into account the H3K9 demethylation activity of Epe1 to be consistent with our experiments[31]. More experiments are necessary to determine which is the dominant mechanism of the enhancement of the repeat-induced RNAi by Epe1.

Liquid-liquid phase separation (LLPS) has been thought to be the principle of the assembly of biological condensates[57,58]. Architectural RNAs (arcRNAs) are a class of RNAs that are essential to the assembly of some condensates[59]. The LLPS is driven by the multi-valent interactions between RNA-binding proteins (RBPs) bound to arcRNAs. RBPs bound to an arcRNA are constrained along the chain from the thermal diffusion of RBPs and this enhances the phase separation[60], much like the Flory-Huggins mechanism of the phase separation of polymers[32]. The latter mechanism is effective for relatively long RNAs. In the assembly of heterochromatin, nascent RNAs produced by transcription are cut into small pieces. Small RNAs play the opposite role: the thermal diffusion of small RNAs allows the long-range interaction between separate regions of chromatin[31,61]. Cells regulate the length of RNAs to achieve different functions. The repeat sequence amplifies the long-range interaction via steady production of small RNAs. The gene silencing through small RNAs is not limited to fission yeast, but is also found in other organisms[37–39]. Our theory can be extended to understand the mechanism of the gene silencing in these organisms.

## Methods

### Kinetics of transcription
The kinetic equations of the state transition have the forms

$$\frac{dn_{on}}{dt} = k_{on}^{p}\rho n_{off} - k_{off}^{p}n_{on} - k_{ini}n_{on} \tag{13}$$

$$\frac{dn_{off}}{dt} = k_{off}^{p}n_{on} - k_{on}^{p}\rho n_{off} + \frac{n_{elo}}{\tau_{elo}} \tag{14}$$

$$\frac{dn_{elo}}{dt} = k_{ini}n_{on} - \frac{n_{elo}}{\tau_{elo}} \tag{15}$$

where $n_{on}$, $n_{off}$, and $n_{elo}$ are the fraction of genes in the bound, unbound, and elongation states, respectively ($n_{on} + n_{off} + n_{elo} = 1$). The state transition of the genes is driven by the binding of a Pol II to the promoter from the nucleosol (the first term of Eq. (13) and the second term of Eq. (14)), the unbinding of the Pol II from the promoter to the nucleosol (the second term of Eq. (13) and the first term of Eq. (14)), the initiation of the transcriptional elongation (the third term of Eq. (13) and the first term of Eq. (15)), and the termination of the transcriptional elongation (the third term of Eq. (14) and the second term of Eq. (15)). $\rho$ is the local concentration of Pol II. $k_{ini}$ is the initiation rate and $\tau_{elo}$ is the elongation time. $k_{on}^{p}$ and $k_{off}^{p}$ are the rate constants that account for the binding and unbinding of Pol IIs to/from the promoter, respectively.

In general, the binding and unbinding of Pol IIs to the promoter is faster than the initiation of the transcription. In the relevant time scale, the binding and unbinding dynamics of Pol IIs reach the chemical equilibrium, $k_{on}^{p}\rho n_{off} - k_{off}^{p}n_{on} = 0$. This situation is analogous to the Michaelis-Menten kinetics, where Pol II acts as "enzyme" and the promoter acts as "substrate". The equilibrium condition leads to the form

$$n_{on} = \frac{\rho}{\rho + K_{p}}n_0 \tag{16}$$

where $n_0(= n_{on} + n_{off} = 1 - n_{elo})$ is the fraction of genes in the bound or unbound state. $K_p(= k_{off}^{p}/k_{on}^{p})$ is the equilibrium constant that accounts for the binding and unbinding of Pol IIs to the promoter. Substituting Eq. (16) into Eq. (15) leads to a kinetic equation only of the fraction $n_{elo}$. In the steady state, the fraction $n_{elo}$ has the form of Eq. (1).

### Free energy of heterochromatin
The free energy of a subchain composed of $g$ units has the form

$$F = F_{ela} + F_{mix} + F_{int} - \mu\frac{\xi^3}{b^3}\left(\rho + \phi_c(n_{on} + n_{elo})\right) + \Pi_{osm}\xi^3 \tag{17}$$

This free energy is the sum of the elastic free energy $F_{ela}$ of the subsection, the free energy $F_{mix}$ due to the mixing entropy, and the free energy $F_{int}$ due to the interaction between units. The fourth and fifth terms of Eq. (17) are the contributions of chemical potential $\mu$, which represents the exchange of Pol II with the exterior (nucleosol), and osmotic pressure $\Pi_{osm}$, which represents the mechanical balance with the exterior. We derive the size $\xi$ of the subchain and the volume fraction $\rho$ of Pol II in the volume pervaded by the subchain by using the free energy.

The elastic free energy has the form

$$\frac{F_{ela}}{k_B T} = \frac{3}{2}\frac{\xi^2}{gb^2} + \frac{3}{2}\frac{gb^2}{\xi^2} \tag{18}$$

Because the elasticity of a polymer results from the conformational entropy, the free energy scales as the thermal energy $k_B T$, where $k_B$ is the Boltzmann constant and $T$ is the absolute temperature. The first term is the elastic free energy due to the stretching of the subsection and the second term is the elastic free energy due to the compression of the subsection.

The mixing free energy has the form

$$\frac{F_{mix}}{k_B T} = \frac{\xi^3}{b^3}\left[\rho\log\rho + \left(1 - \rho - \frac{b^3 g}{\xi^3}\right)\log\left(1 - \rho - \frac{b^3 g}{\xi^3}\right)\right] \tag{19}$$

The first term of Eq. (19) is the free energy due to the mixing entropy of Pol II and the second term of this equation is the free energy due to the mixing entropy of solvent. For simplicity, we neglected the excluded volume of Pol IIs that are bound to the chromatin units because it is not essential to the physics of heterochromatin assembly and increases the number of unknown parameters, see also Supplementary Note 3 and Supplementary Figure 15.

The interaction free energy has the form

$$\frac{F_{int}}{k_B T} = -\frac{\xi^3}{b^3}\chi\sigma^2\left(\frac{b^3 g}{\xi^3}\right)^2 - \frac{\xi^3}{b^3}\chi_0\left(\frac{b^3 g}{\xi^3}\right)^2 \tag{20}$$

The free energy is a function of the size $\xi$ of the subchain and the volume fraction $\rho$ of Pol IIs. The first derivatives of the free energy with respect to $\xi$ and $\rho$ lead to the balance of the osmotic pressure (the equation of state) and the equality of chemical potential of Pol IIs between the subchain and the exterior. Equation (9) is derived by rewriting the equation of state with the chromatin volume fraction

$$\phi_c = \frac{b^3 g}{\xi^3} \tag{21}$$

Equation (10) is derived by rewriting the equality of chemical potentials by using $\rho_0 = 1/(1 + e^{-\mu/(k_B T)})$.

### Diffusion of small RNA
Small RNAs produced by RDRC/Dicers diffuse away from the nuclear membrane. The local concentration $c(z,t)$ of small RNAs follow the diffusion equation

$$\frac{\partial}{\partial t}c(z,t) = D\frac{\partial^2}{\partial z^2}c(z,t) - k_d c(z,t) \tag{22}$$

where $D$ is the diffusivity of small RNAs and $k_d$ is the degradation rate of small RNAs. $z$ is the coordinate from the surface of the nuclear membrane, see Fig. 2. The boundary condition of the local concentration $c(z,t)$ has the form

$$-D\frac{\partial}{\partial z}c(z,t)\bigg|_{z\to 0} = S \tag{23}$$

where the rate $S$ of small RNA production is given by Eq. (7). Equation (4) is derived by solving Eq. (22) in the steady state and use Eq. (23) to determine the integral constant.

### Reporting summary
Further information on research design is available in the Nature Portfolio Reporting Summary linked to this article.

## Data availability
The data obtained by the numerical calculations are available in figshare with the identier (https://doi.org/10.6084/m9.figshare.22098833).

## Code availability
The Mathematica file used for the numerical calculations are available in figshare with the identier (https://doi.org/10.6084/m9.figshare.22098833).

# ARTICLE

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

## Acknowledgements
This work was supported by KAKENHI to T.Y. Grant Numbers 20H05934 (Grant-In-
Aid for Transformative Research A, "Genome modality", MEXT), 21H00241 (Grant-In-
Aid for Scientific Research on Innovative Area, "Chromatin potential", MEXT),
21K03479 (Grant-In-Aid for Scientific Research (C), JSPS), to Y.M. Grant Numbers
12206045 (Grant-in-Aid for Scientific Research on Priority Areas, MEXT) and 21247001
(Grant-in-Aid for Scientific Research (A), JSPS).

## Author contributions
T.Y., T.A., and Y.M. contributed to the conceptualization of the research and wrote/
revised the manuscript. T.Y. constructed the theoretical model and analyzed it.

## Competing interests
The authors declare no competing interests.

## Additional information
**Supplementary information** The online version contains supplementary material
available at https://doi.org/10.1038/s42003-023-05154-w.

**Peer review information** *Communications Biology* thanks Mina Farag and the other,
anonymous, reviewer(s) for their contribution to the peer review of this work. Primary
Handling Editors: Quan-Xing Liu and Gene Chong. A peer review file is available.

**Publisher's note** Springer Nature remains neutral with regard to jurisdictional claims in
published maps and institutional affiliations.

