## [Peer Review File · Communications Biology]

Reviewers' comments:

Reviewer #1 (Remarks to the Author):

In this manuscript, the authors combine the approaches in systems biology and polymer physics to provide a modeling framework for the assembly of constitutive heterochromatin in fission yeast with considering the processes of small RNA production and histone methylation. They reveal the role of the polymeric nature of tandemly repeated genes to enhance the assembly of constitutive heterochromatin. Their findings are of much importance to improve our understanding about the assembly of heterochromatin in cells. Therefore, I would like to recommend it for publication in Communications Biology after the following problems are addressed.

It feels confused that '... is derived by the minimization of the free energy in the spirit of the Flory theory' without showing the minimizing of the free energy given by equation (M5). Please add it or make other further explanations. And more importantly, please explain why to use the free energy to obtain the volume fraction ϕ_C and why it is reasonable and effective.

And other details:

1. The used 'the PDF Appendix' in Lines 150, 212 and 741 should be replaced by 'SI'.
2. 'probably p ' should be 'probability p ' in Line 186.
3. It should provide corresponding supports or results for the state "the general feature of our results does not depend on these values" in Lines 218 and 304.
4. Delete ')' in Line 229 after 330kb.
5. ' k_{on} ' in Line 234 should be ' k_{ini} '.
6. Please explain how to determine the critical value N_c in Line 237.
7. Please check 'Eq.(4)' in Lines 282, 295 and 523 should be 'Eq.(9)'.
8. Please check the sentence 'Indeed, the production rate ..., see Fig.6b and c' in Lines 321-323.
9. Please check the sentence 'The production rate of small RNAs ... in Fig.7b and c' in Lines 338-341.
10. It is better to add the meaning of z in equation (M9) in Line 529.
11. I feel confuse for the orange color of RDRC/Dicers used in Fig.2a. And, it is better to indicate what a promoter is in Fig.2b.
12. Recorrect the type of $\sigma_{q_{on}}$ and τ_{elo} in the caption in Fig.5.

Reviewer #2 (Remarks to the Author):

In their paper "Polymeric nature of tandemly repeated genes enhances assembly of constitutive heterochromatin in fission yeast," Yamamoto, Asanuma, and Murakami develop a kinetic model to understand the relationship between heterochromatin assembly, small RNA production, and histone methylation in fission yeast. This paper is a follow-up to an experimental paper published by many of the same authors. As I understand them, the main results and predictions set forth by this paper are as follows:

1. The repeated polymeric nature of centromeric genes promotes the binding of nascent RNAs to RDRC/Dicers at the surface of the nuclear membrane.
2. Chromatin length and transcriptional elongation time are both important factors in determining whether a genomic region is euchromatin-like or heterochromatin-like.
3. Swi6-induced gene compaction decreases the local concentration of Pol II and decreases transcription.
4. If the Swi6-induced gene compaction is strong enough, then decreasing methylation leads to even more chain compaction, less Pol II, and decreased transcription. This is a potential explanation of their results in their prior paper. Notably, considering Epe1 as only a transcriptional activator and not as a demethylase causes their model to disagree with their experimental results. This suggests that at the

very least the role of Epe1 as a demethylase is important. However, the importance of the role of Epe1 as a transcriptional activator is unclear.

Taken together, I think these results and predictions would be of general interest to the field. In particular, the connections to polymer compaction / expansion are highly interesting and, as the authors mention, would be worth validating experimentally in the future. Below, I outline my comments and questions that the authors may choose to account for during the revision process.

1. The authors motivate this work by contrasting this mechanism with phase separation. Specifically, they say that “The nucleus of a fission yeast has only three chromosomes, each having a centromeric region of 40 - 110 kbps (which are estimated to be 24 - 67 Kuhn units). Because phase separation is a collective phenomenon of many polymers, the heterochromatin of fission yeast is not likely to be assembled by the phase separation of chromatin.” I would push back on this assertion. Even one chain can undergo phase separation if it is exceedingly long and highly multivalent. While the authors mention that the number of Kuhn segments is relatively small, the actual length of the chain is quite large — significantly larger than any proteins that undergo phase separation. Thus, I would suggest that the authors either include stronger evidence that these chains do not undergo phase separation or remove this statement altogether and state that they are suggesting an alternative mechanism of heterochromatin organization.

2. I am not an expert in heterochromatin organization, so please excuse my ignorance here. I found it slightly unclear exactly what the authors are proposing as the relationship between the tandem repeat nature of the centromeric genes and the binding of nascent RNAs to RDRC/Dicers. I recognize that they are drawing comparisons to surface adhesion of polymers as in Fig. 1, which makes sense. However, is it that the polymeric nature of the genes themselves promote binding to a specific surface? Or is the polymeric nature of the nascent RNAs promoting binding? Furthermore, is the surface in this case the surface of RDRC/Dicer molecules or the surface of the nuclear membrane? Based on the results of Figure 5, it seems that the genes are binding RDRC/Dicers, but I think these points could use some clarification early on for those less familiar with the prior work. I would suggest reworking Figure 2 to make the whole model clearer.

3. As a modeling paper, this work includes numerous symbols and parameters. I appreciate that the authors include two tables to describe the parameter values. However, I often found myself needing to look at prior paragraphs to recall what each symbol stands for. As such, I think it would be worthwhile to include a glossary that defines all of the unique symbols used throughout this work. This would be a useful reference for the reader.

4. In Figure 4, τ_{sp1} and τ_{sp2} should be defined in the figure caption.

5. Also in Figure 4, the magenta-colored dashed curve and dotted lines were hard to differentiate, which led to some difficulty in reading the graph at first. I recommend using a different method to demarcate τ_{sp1} and τ_{sp2} .

6. In Figure 5, the differently-colored curves typically overlay each other in the regime of long elongation times. Thus, saying that “The average degree of H3K9 methylation and the average production rate of small RNAs also increase dramatically with increasing the number N of genes” seems misleading, since this is only true for certain values of the elongation time.

7. The legend for Table 2 states “The volume and concentration of Pol II are estimated to be $6 \times 10^3 \text{ nm}^2$ (Spahr et al. 2009) and 50 (the number of Pol II per cell is 3×10^4 (Borggreffe et al. 2001) and the size of a nucleus of fission yeast is in the order of $1 \mu\text{m}^3$ (Wang et al. 2016)), respectively.” The volume units are nm^2 , which seems odd, and the concentration is unitless. I believe these both need to be fixed.

Reply to Reviewer #1

In this manuscript, the authors combine the approaches in systems biology and polymer physics to provide a modeling framework for the assembly of constitutive heterochromatin in fission yeast with considering the processes of small RNA production and histone methylation. They reveal the role of the polymeric nature of tandemly repeated genes to enhance the assembly of constitutive heterochromatin. Their findings are of much importance to improve our understanding about the assembly of heterochromatin in cells. Therefore, I would like to recommend it for publication in Communications Biology after the following problems are addressed.

Thank you very much for your recommendation and constructive comments. Your comments were useful to make our manuscript more concise and to make it more accessible to wider audience. We thus revised our manuscript in line with your suggestions. The revised parts are highlighted by red letters. We also added Supplementary Note 2 and 3 and Supplementary Figures 6 - 15 in the Supplementary Information. Our point-by-point reply follows.

It feels confused that ‘... is derived by the minimization of the free energy in the spirit of the Flory theory’ without showing the minimizing of the free energy given by equation (M5). Please add it or make other further explanations.

We summarized the derivation of eq. (9) from the minimization of the free energy (eq. (17)) in the Supplementary Note 3.

And more importantly, please explain why to use the free energy to obtain the volume fraction ϕ_C and why it is reasonable and effective.

Thank you very much for this comment. The heterochromatin of fission yeasts is not in the thermodynamic equilibrium. It is therefore necessary to clearly explain why our approach is effective. First, the equation of state that relates the osmotic pressure and the volume fraction of chromatin is derived by the thermodynamic relationship. Our use of the free energy minimization by introducing the contribution of osmotic pressure is just for convenience. To avoid misunderstanding, we revised the sentence in L269-L275:

Eq. (9) is an extension of the equation of state that predicts the coil-globule transition of polymers and is derived by using the free energy in the spirit of the Flory theory⁵⁰, see also Fig. 3 and eq. (17) in Materials and Methods (the detail of the derivation is shown in Supplementary Note 3).

Second, to specify cases in which the balance of stresses in eq. (9) is applicable, we also added a sentence in the same paragraph:

This treatment is effective for cases in which the binding and unbinding of the units is rate limited: in such cases, the polymer dynamics associated with the binding and unbinding of the units is negligible and thus the local equilibrium approximation is applicable in the length scale of the subchain.

1. The used 'the PDF Appendix' in Lines 150, 212 and 741 should be replaced by 'SI'.
2. 'probably p ' should be 'probability p ' in Line 186.
4. Delete ')' in Line 229 after 330kb.
7. Please check 'Eq.(4)' in Lines 282, 295 and 523 should be 'Eq.(9)'.
12. Recorrect the type of σ_{on} and τ_{elo} in the caption in Fig.5.

These typos were corrected. Thank you for noticing them.

3. It should provide corresponding supports or results for the state "the general feature of our results does not depend on these values" in Lines 218 and 304.

The binding probably p is the main parameter to describe the system. We thus performed numerical calculations of p with different values of parameters. The results are shown in Supplementary Figures 6 - 15.

5. ' k_{on} ' in Line 234 should be ' k_{ini} '.
6. Please explain how to determine the critical value N_c in Line 237.

We added Supplementary Note 2 to show the derivation of Fig. 4, including two limits of instability, τ_{sp1} and τ_{sp2} , and the critical value N_c . We added a sentence in the caption of Fig. 4 to notify it:

The derivation of this figure is shown in Supplementary Note 2.

We noticed that the labels of figs. 4 and 5 was wrong, rather than the text. We thus corrected these labels.

8. Please check the sentence 'Indeed, the production rate ..., see Fig.6b and c' in Lines 321-323.

To make the description more precise, we revised the corresponding sentence (L310-L312) as follows:

Indeed, the production rate $S\xi_N^2$ of small RNAs and the degree σ of H3K9 methylation in the bound state decreases with increasing the interaction parameter χ for the regime of long elongation time, see Fig. 6**b** and **c**.

9. Please check the sentence 'The production rate of small RNAs ... in Fig.7b and c' in Lines 338-341.

To make the description more precise, we revised the corresponding sentence (L325-L330) as follows:

It implies that if the inverse demethylation rate k_{dm}^{-1} is larger than the maximum in the wild type, the binding probability p decreases with decreasing the rate k_{dm} (which corresponds to the depletion of Epe1). In this regime, the production rate of small RNAs in the bound state also decreases with decreasing the demethylation rate k_{dm} , while the degree of H3K9 methylation increases with decreasing the demethylation rate k_{dm} , see the orange and magenta lines in Fig. 7**b** and **c**.

10. It is better to add the meaning of z in equation (M9) in Line 529.

We added the following sentence at L518 below eq. (21) (eq. (M9) in the previous version) to address the meaning of z :

z is the coordinate from the surface of the nuclear membrane, see Fig. 2.

11. I feel confuse for the orange color of RDRC/Dicers used in Fig.2a. And, it is better to indicate what a promoter is in Fig.2b.

We changed the color of RDRC/Dicers in Fig. 2a to black and indicated promoter in Fig. 2b.

Reply to Reviewer #2

In their paper “Polymeric nature of tandemly repeated genes enhances assembly of constitutive heterochromatin in fission yeast,” Yamamoto, Asanuma, and Murakami develop a kinetic model to understand the relationship between heterochromatin assembly, small RNA production, and histone methylation in fission yeast. This paper is a follow-up to an experimental paper published by many of the same authors. As I understand them, the main results and predictions set forth by this paper are as follows:

- 1. The repeated polymeric nature of centromeric genes promotes the binding of nascent RNAs to RDRC/Dicers at the surface of the nuclear membrane.*
- 2. Chromatin length and transcriptional elongation time are both important factors in determining whether a genomic region is euchromatin-like or heterochromatin-like.*
- 3. Swi6-induced gene compaction decreases the local concentration of Pol II and decreases transcription.*
- 4. If the Swi6-induced gene compaction is strong enough, then decreasing methylation leads to even more chain compaction, less Pol II, and decreased transcription. This is a potential explanation of their results in their prior paper. Notably, considering Epe1 as only a transcriptional activator and not as a demethylase causes their model to disagree with their experimental results. This suggests that at the very least the role of Epe1 as a demethylase is important. However, the importance of the role of Epe1 as a transcriptional activator is*

unclear.

Taken together, I think these results and predictions would be of general interest to the field. In particular, the connections to polymer compaction / expansion are highly interesting and, as the authors mention, would be worth validating experimentally in the future. Below, I outline my comments and questions that the authors may choose to account for during the revision process.

Thank you very much for your recommendation and constructive comments. Your comments are useful to make our manuscript more concise and make it accessible to wider audience. We thus revised our manuscript in line with your suggestions. The revised parts are highlighted by red letters. Our point-by-point reply follows:

1. The authors motivate this work by contrasting this mechanism with phase separation. Specifically, they say that “The nucleus of a fission yeast has only three chromosomes, each having a centromeric region of 40 - 110 kbps (which are estimated to be 24 - 67 Kuhn units). Because phase separation is a collective phenomenon of many polymers, the heterochromatin of fission yeast is not likely to be assembled by the phase separation of chromatin.” I would push back on this assertion. Even one chain can undergo phase separation if it is exceedingly long and highly multivalent. While the authors mention that the number of Kuhn segments is relatively small, the actual length of the chain is quite large — significantly larger than any proteins that undergo phase separation. Thus, I would suggest that the authors either include stronger evidence that these chains do not undergo phase separation or remove this statement altogether and state that they are suggesting an alternative mechanism of heterochromatin organization.

In line with your suggestion, we removed the statement altogether from both abstract and introduction.

2. I am not an expert in heterochromatin organization, so please excuse my ignorance here. I found it slightly unclear exactly what the authors are proposing as the relationship between the tandem repeat nature of the centromeric genes and the binding of nascent RNAs to RDRC/Dicers. I recognize that they are drawing comparisons to surface adhesion of polymers as in Fig. 1, which makes sense. However, is it that the polymeric nature of the genes themselves promote binding to a specific surface? Or is the polymeric nature of the nascent RNAs promoting binding? Furthermore, is the surface in this case the surface of RDRC/Dicer molecules or the surface of the nuclear membrane? Based on the results of Figure 5, it seems that the genes are binding RDRC/Dicers, but I think these points could

use some clarification early on for those less familiar with the prior work. I would suggest reworking Figure 2 to make the whole model clearer.

We added thick cyan and black lines that highlight the connectivity of genes along chromatin and the 'connectivity' due to the localization of RDRC/Dicers at the surface of nuclear membrane in Figure 2:

We also added the following sentence (L706-L709) in the caption of Fig. 2 to describe the meaning of these thick lines:

If more than one gene are bound to a RDRC/Dicer, unbound genes in the tandem repeat are also localized at the vicinity of RDRC/Dicers because the genes are connected through the chromatin and RDRC/Dicers are localized at the surface of the nuclear membrane, see the thick cyan and black lines.

To avoid mistake between the surfaces of nuclear membranes and the surfaces RDRC/Dicers, we make sure to write “the surface of the nuclear membrane”, not just “surface”. To avoid mistake between the polymeric nature of chromatin and the polymeric nature of nascent RNAs, we added the following sentence at L118-L121:

Nascent RNAs produced by the transcription of heterochromatin regions are retained to the chromatin via RITS complexes^{28,42,43}. A complex of the chromatin unit and nascent RNA can be thus viewed as one unit that can bind to RDRC/Dicers at the surface of the nuclear membrane.

We also revised Fig. 2 to make clear that nascent RNAs are retained to chromatin by RITS complexes (if the chromatin is H3K9 methylated).

To address the punchline of this paper clearly, we also revised a sentence on L395-L400:

The stable binding is promoted by the localization of nascent RNAs along DNA and of RDRC/Dicers on the surface of the nuclear membrane with a mechanism analogous to the surface adhesion of polymers: if a nascent RNA produced from a gene is bound to a RDRC/Dicer, other nascent RNAs in the tandem repeat are at the vicinity to the surface, where other RDRC/Dicers are localized.

3. As a modeling paper, this work includes numerous symbols and parameters. I appreciate that the authors include two tables to describe the parameter values. However, I often found myself needing to look at prior paragraphs to recall what each symbol stands for. As such, I think it would be worthwhile to include a glossary that defines all of the unique symbols used throughout this work. This would be a useful reference for the reader.

Thank you for this idea. We added a glossary of symbols in Supplementary Table 2.

4. In Figure 4, τ_{sp1} and τ_{sp2} should be defined in the figure caption.

To clarify the definition of τ_{sp1} and τ_{sp2} , we added the following sentence in the caption of Figure 4 (L723-L727):

The stable solutions are shown by the solid lines and the unstable solution is shown by the broken line. τ_{sp1} and τ_{sp2} are the values of the elongation time at which the heterochromatin and euchromatin solutions become unstable, respectively, see the light green dotted lines.

5. Also in Figure 4, the magenta-colored dashed curve and dotted lines were hard to differentiate, which led to some difficulty in reading the graph at first. I recommend using a different method to demarcate τ_{sp1} and τ_{sp2} .

We used green color (instead of magenta) dotted lines to demarcate τ_{sp1} and τ_{sp2} .

See also the revision of the figure caption described in the previous comment for the explanation of the green dotted lines.

6. In Figure 5, the differently-colored curves typically overlay each other in the regime of long elongation times. Thus, saying that “The average degree σq_{on} of H3K9 methylation and the average production rate of small RNAs also increase dramatically with increasing the number N of genes” seems misleading, since this is only true for certain values of the elongation time.

To make the description more precise, we revised the corresponding sentence on L244-L249:

The average degree σq_{on} of H3K9 methylation and the average production rate of small RNAs increase with increasing the number N of genes and eventually saturate for the regime of long elongation time, reflecting the feature of q_{on} , see Fig. 5b and c. The increase of the level of H3K9 methylation and small RNAs with the number N of genes in the tandem repeat is consistent with our recent experiments³¹.

We also revised another sentence, L237-L240, that explains Fig. 5a:

The probability q_{on} of the repeat at the bound state increases as the elongation time τ_{elo} increases and eventually becomes $q_{on} \approx 1$, where the repeat is stably bound to the surface of nuclear membranes, see Fig. 5a. Notably, the width of the window of elongation time at which $q_{on} \approx 1$ increases as the number N of genes increases.

7. The legend for Table 2 states “The volume and concentration of Pol II are estimated to be $6 \times 10^3 \text{ nm}^2$ (Spahr et al. 2009) and 50 (the number of Pol II per cell is 3×10^4 (Borggreve et al. 2001) and the size of a nucleus of fission yeast is in the order of $1 \mu\text{m}^3$ (Wang et al. 2016)), respectively.” The volume units are nm^2 , which seems odd, and the concentration is unitless. I believe these both need to be fixed.

The missing units were corrected.

REVIEWERS' COMMENTS:

Reviewer #2 (Remarks to the Author):

The authors have addressed all of my prior suggestions fully and I have no further comments or recommendations.